# A Non-Contact Heart Rate Measurement Method Resistant to Illumination Changes Based on Fast Wavelet Transform and Second-Order Blind Identification in Far-Field Environments

Rui Yuan*
Anhui Province Key Laboratory of
Multimodal Cognitive Computation,
School of Computer Science and
Technology
Anhui University
Hefei, China

Chao Zhang†
Anhui Province Key Laboratory of
Multimodal Cognitive Computation,
School of Computer Science and
Technology
Anhui University
Hefei, China

## ABSTRACT

Heart rate(HR) is a key parameter for evaluating a person's physiological condition. In recent years, there have been many researches on remote heart rate measurement. However, these methods are mostly conducted in close-range scenarios, making them inapplicable in many scenarios. Remote photoplethysmography (rPPG) provides more possibilities for heart rate measurement in far-field environments. Moreover,the performance of heart rate measurement will be significantly reduced when the subject's movement and the illumination changing. We propose a rPPG framework for heart rate detection, which selects a larger region of interest (ROI) using feature point tracking in far-field environments. The combination of fast wavelet transform (FWT) and second-order blind identification (SOBI) is used to resist illumination interference and most of the motion interference. Singular spectrum analysis (SSA) is then used to resist residual motion interference.In addition, we collected a database of illumination changes in far-field environments and tested our framework with it. The results show that our method is superior to all previous methods.

**Index Terms:** Heart rate(HR),Far-field environments,Fast wavelet transform(FWT),Second-order blind identification(SOBI).

## 1 INTRODUCTION

Heart rate is a key parameter for evaluating a person's physiological condition. Early symptoms of cardiovascular disease are not easy to detect and require specific heart rate monitoring equipment, such as electrocardiogram (ECG), which must be in contact with the surface of the skin. If continuous long-term monitoring is required, it can cause inconvenience to the user or patient. In addition, ECG devices are often large in size and expensive, resulting in high measurement costs, which are not conducive to real-time monitoring of the user's physiological and psychological health conditions.People urgently hope to be able to learn their cardiovascular physiological condition through safe and convenient means. Compared with contact-based devices, video-based non-contact heart rate monitoring has obvious advantages. This method uses a consumer level camera to sense the heart rate by capturing body surface videos, so it is very cheap and user friendly.The non-contact method overcomes the disadvantages of contact based heart rate measurement and is therefore widely used in human-computer interaction, health monitoring, and other fields.

*e-mail: E21301281@stu.ahu.edu.cn
†e-mail: iiphci_ahu@163.com

rPPG is a non-contact method for measuring heart rate. rPPG technology has broad application prospects, for example, in scenarios such as neonates, burn patients, or long-term monitoring, where measurement does not require a far distance. Currently, the near-distance rPPG technology has achieved good application effects in these scenes.However, in some environments where heart rate measurement needs to be performed at a considerable distance, most of the current rPPG applications are unable to meet the requirements.For example, in scenarios such as court hearings, live sports events,and online interviews, it is necessary to obtain the heart rate of individuals over a certain distance.Previous research has shown the feasibility of video-based heart rate measurement, but in real-world environments, changes of illumination and human motion can significantly affect measurement results. It is difficult to avoid interference from illumination during long-term heart rate monitoring, as changes in illumination include various forms of noise caused by environmental changes,such as flickering indoor illumination or changes in outdoor natural indoor illumination.Additionally, it is difficult to avoid interference from human motion,which includes both rigid movements such as head tilting and nonrigid movements such as blinking and smiling. In this paper, we propose a framework that can effectively resist these interferences in a far-field environment. Furthermore, we collected a database of illumination changes in far-field environments and used this database to test our algorithm.

## 2 RELATED WORKS

rPPG is a non-contact method for measuring heart rate from a distance using a camera. The basic rPPG process involves the following steps: First,capturing a video of the subject's face using a camera with sufficient resolution and frame rate, Then,selecting an ROI on the subject's face, typically around the forehead or cheek. After that,extracting the BVP (Blood Volume Pulse) signal from the selected ROI using various signal processing techniques, processing the BVP signal to remove any noise or artifacts.Finally,estimating the heart rate from the processed BVP signal.

Verkruysse et al. [12] first proposed the use of a regular high-definition camera with rPPG technology to measure heart rate. In ideal conditions, they used the G-BVP method to estimate heart rate and achieved relatively accurate results. However, in practical scenarios, rPPG technology struggles to extract accurate BVP waveforms due to changes of illumination and significant motion. To address these issues, Poh et al. [9] first proposed a method based on independent component analysis (ICA). They believed that the R, G, and B channel signals from the imaging device were mixed with BVP signals and noise signals and that using ICA separation methods could isolate the BVP signals from the three channels. The results showed that the ICA separation method provided more accurate results than using the green channel alone. Lewandowska

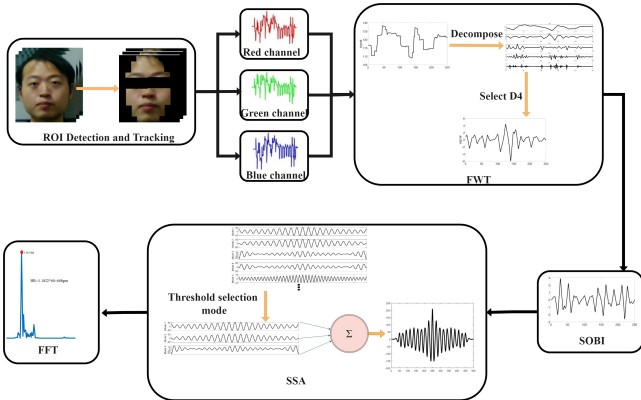

Figure 1: Proposed framework for heart rate measurement.

et al. [7] used principal component analysis (PCA) to select the strongest periodic signals as BVP signals, resulting in accurate heart rate measurements. Li et al. [8] proposed a method based on face tracking and normalized least mean square adaptive filtering to combat the effects of illumination and motion. Chen et al. [3] used joint blind source separation and empirical mode decomposition to analyze color signals from multiple facial subregions to resist the effects of illumination changes. In addition, there are also some model-based methods [4] [14], which believe that motion artifacts can be eliminated by linear combinations of the R, G, and B channels. When detecting heart rate based on rPPG, reliable region of interest (ROI) detection and tracking are key steps. By removing areas on the face that are more susceptible to motion or change and using local motion compensation methods [9], the accuracy of heart rate measurement can be ensured.

The above methods aim to mitigate the effects of illumination changes and human motion on heart rate measurement as much as possible. However, their experiments were carried out with the distance between camera and the subjects very close, which imposes many limitations on the practical application scenarios. Al-Naji et al. [1] used a framework combining video magnification and blind source separation to reduce the impact of illumination changes on heart rate measurement. They increased the distance of heart rate measurement and provided more space for the application of rPPG. Since the facial image is smaller in a far-field environment, they selected the entire face as the ROI. To avoid interference from non-rigid motion on the measurement results, they removed the eye region for heart rate measurement as accurately as possible. However, their experiments only involved six individuals, which may lead to insufficient data to prove the effectiveness of the experimental measurement. In addition, the ROI they selected includes some background outside the face, which may obtain some non-physiological signals of the person during the video signal processing, resulting in inaccurate heart rate measurement. Moreover, the region they selected when removing the human eyes is too large, which will continue to amplify the disadvantage of the image being too small in the far-field environment, resulting in a ROI selected for heart rate measurement that is too small and inaccurate results.

## 3 A FRAMEWORK FOR RESISTING ILLUMINATION AND MOTION INTERFERENCE IN FAR-FIELD ENVIRONMENTS

As there is no publicly available database for illumination changes in far-field environments, we collected a database specifically for this scenario. Additionally, we propose a framework that can resist illumination and slight motion interference in far-field environments.Our framework consists of three steps, as shown in Fig. 1. In the first step, we need to obtain the ROI that contains the raw

physiological signal of the person. We use the Viola-Jones(VJ) algorithm [13] to detect the face in the first frame of the image and then use the Kanade-Lucas-Tomasi (KLT) algorithm [11] to track the position of the ROI. In each frame, we convert the ROI image into an RGB three-channel signal by spatial averaging. The purpose of the second step is to reduce the interference caused by changes in illumination. We perform fast wavelet transform (FWT) [15] on the three-channel signal, preprocess the signal to remove some of the interference of illumination, and then use the SOBI algorithm [2] to process the signal to remove both illumination and motion interference. The purpose of the third step is to filter out residual motion interference. We perform singular spectrum analysis (SSA) [5] on the processed signal, which can resist motion interference to some extent. Then, we estimate the heart rate using Fourier transform. The details of each step will be explained in the following sections.

### 3.1 ROI Detection and Tracking

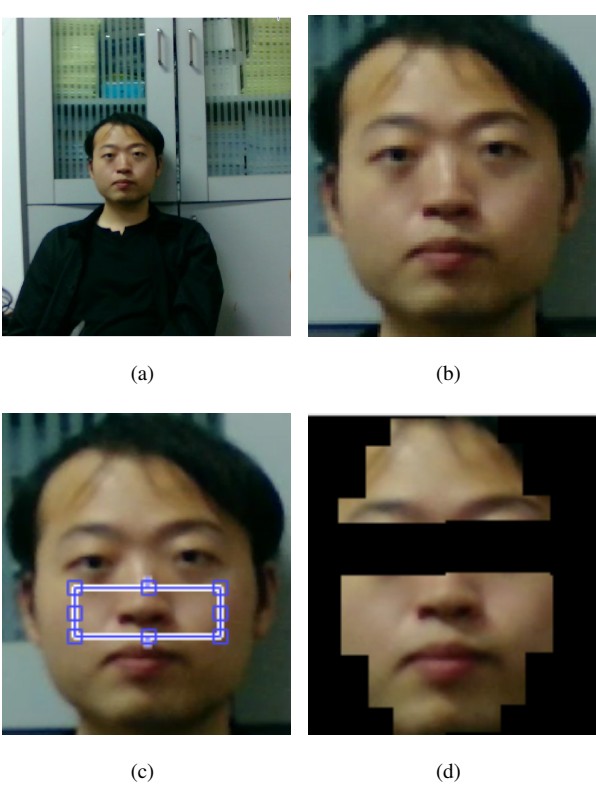

(a)        (b)

(c)        (d)

Figure 2: Video frame. (a) A frame from the video, (b) Facial region image, (c) Image with generally selected ROI, (d) Image with feature point tracked ROI.

The selection of the face region of interest (ROI) is a crucial step in rPPG heart rate measurement. Figure 2(a) shows a frame from one of the videos in our self-collected dataset, captured under far-field conditions with illumination changes.Fig. 2(b) shows an image without the selected ROI. Typically, as shown in Fig. 2(c), the region below the eyes and above the mouth is generally selected as the ROI because this area is not susceptible to motion interference and contains dense capillaries that provide the required signal, making it a good choice for ROI. However, if a fixed box in the video frame is used to represent the ROI, the box may deviate from the original ROI area result in losing the area for obtaining physiological signals. In a far-field environment, since the facial area is relatively small, our goal is to include as much of the facial area as possible in the ROI, while excluding the eye region, which produces non-rigid motion

and interferes to heart rate measurement.As shown in Fig. 2(d),we used the Viola-Jones face detector [13] to detect the face in the first frame, which provides a rectangular box containing the approximate position of the face. We used a rough facial template to locate the skin areas above and below the eyes and remove the eye region. To remove the background, we detected feature points [10] using the Minimum Eigenvalue Algorithm and selected suitable facial landmarks within the rectangular box to include as much of the facial area as possible in the ROI. Then, we used the Kanade-Lucas-Tomasi (KLT) technique [11] to track the face in each frame of the video. By tracking the feature points in the current and next frames, we adjusted the spatial orientation and size of the ROI in 2D and obtained the raw RGB signal by spatially averaging the pixel intensity values in each frame's ROI.

### 3.2 Illumination Rectification

In this section, we aim to remove illumination changes as much as possible. To achieve this, we use wavelet transform to process the RGB signal. Wavelet transform is a time-frequency localized analysis method, whose window area is fixed but time and frequency windows are variable. Therefore, wavelet transform has the characteristics of multi-resolution analysis and can represent the local features of signals in both time and frequency domains. In simple terms, wavelet transform can decompose a signal into components of different frequencies, in order to better understand the temporal and spectral characteristics of the signal. In the low-frequency part, wavelet transform has higher spectral resolution and lower time resolution, while in the high-frequency part, it has higher time resolution and lower spectral resolution. These two characteristics are consistent with the characteristics of slow changes in low-frequency signals and rapid changes in high-frequency signals, making wavelet transform adaptive to different types and frequencies of signals. Through fast wavelet transform, we can decompose the influence of illumination changes on the RGB signal into different frequency components, and select appropriate components for filtering to extract the heart rate signal and remove noise. This process is similar to passing the signal through a band-pass filter, retaining only the signals within the target frequency range and filtering out other frequency components. Finally, the filtered components are combined into a clean heart rate signal, thereby removing the interference of illumination changes on heart rate detection. Therefore, the fast wavelet transform algorithm can be regarded as a filter. For a 1D input signal $f(t)$, its decomposition formula is as follows:

$$f(t) = A_n + D_n + D_{n-1} + ... + D_1 \quad (1)$$

In formula 1, $n$ represents the number of decomposition levels of the signal. Through wavelet decomposition, we can divide the signal into low-frequency and high-frequency parts, represented by $A$ and $D$, respectively [15].. In order to better process the signal, we use filters to separate the high-frequency and low-frequency waves and convert them into frequencies. Then, we recombine them and perform dimensionality reduction to integrate local information (low-frequency) and spatial information (high-frequency). The proposed method adopts 4-level wavelet decomposition and chooses the db3 wavelet type,which has good time-domain and frequency-domain characteristics.The db3 wavelet consists of three scales of wavelet functions and three scales of wavelet packet functions, which can provide a higher signal compression ratio and better signal reconstruction quality. In addition, the db3 wavelet has good performance in both low-frequency and high-frequency decomposition, which can effectively extract heart rate signals while removing noise and interference. Therefore, choosing db3 wavelet for one-dimensional discrete wavelet transformation can improve the measurement accuracy.Considering that the frequency range of heart rate is completely covered by the frequency range of $D_4$, we choose to set the decomposition coefficients of $D_1$, $D_2$, $D_3$, and $A_4$ to zero, and retain

the decomposition coefficient of $D_4$. Then, we reconstruct the decomposition coefficients of $D_1$, $D_2$, $D_3$, $D_4$, and $A_4$ to obtain the preprocessed signal. This preprocessing method can better highlight the characteristics of the heart rate signal and improve the quality of the signal. By further analyzing the preprocessed signal, we can more accurately measure the heart rate.

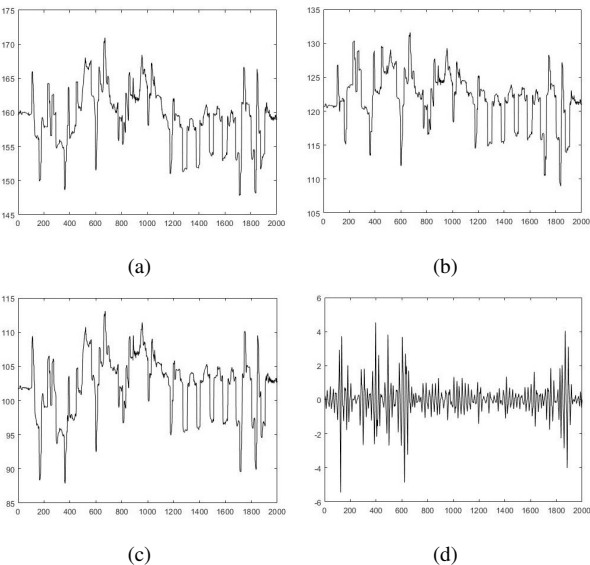

Figure 3: Raw RGB signals and SOBI processed signal. (a) Raw signal of red channel, (b) Raw signal of green channel, (c) Raw signal of blue channel, (d) SOBI processed signal.

The blind source separation algorithm is used to further denoise the preprocessed signal, which can remove most of the lighting and motion noise. We use SOBI for denoising, which is a type of ICA (Independent Component Analysis) algorithm that differs from other ICA algorithms in that it uses second-order statistical information to reconstruct source signal. SOBI method can deal with non-Gaussian noise better, so it is more suitable for BVP signal processing. Therefore, in the case of the interference of illumination and motion interference, the SOBI method is better than other ICA methods. The observed signal $X$ is obtained by linearly mixing the source signals $S$ through the mixing matrix $A$. The source signals can be represented as $S = [s_1, s_2, s_3]^T$, and the mathematical expression is $X = AS$. Both the source signals $S$ and the mixing matrix $A$ are unknown and Formula 3 can be used to separate the observed signal$X$. Where $W$ is the separation matrix, represented as $W = A^{-1}$, and $Y = [y_1, y_2, y_3]^T$ is the estimated value of the source signals $S$. $W$ is randomly initialized and continuously optimized until $Y$ is close to $S$, obtaining the desired signal $X$.

$$x_i = \sum_{j=1}^{3} a_{ij} s_j (1 \leq i \leq 3) \quad (2)$$

$$y_i = \sum_{j=1}^{3} w_{ij} x_j (1 \leq i \leq 3) \quad (3)$$

### 3.3 Motion Elimination

We use SSA to remove residual motion noise from the BVP signal.SSA is an effective method for processing nonlinear time series data, which can decompose time series into meaningful components without prior knowledge. It can directly extract the artifact spectrum from the BVP signal of the facial ROI, assuming that the facial ROI

contains all interference information and that the noise artifact is unrelated to the pulse signal. By applying SSA, dominant noise artifacts can be extracted, and it is found that the effect of noise artifacts on all RGB channels is almost the same. To eliminate residual noise artifacts in the extracted BVP signal, SSA can be applied to estimate the spectrum and obtain a BVP signal without noise artifacts. The modal decomposition calculated using SSA is as follows.Let $x(t)$ be the normalized BVP signal at time $t$, and define its trajectory matrix as follows:

$$X = \begin{bmatrix} x(1) & x(2) & \ldots & x(m) \\ x(2) & x(3) & \ldots & x(m+1) \\ \vdots & \vdots & \ddots & \vdots \\ x(n-m+1) & x(n-m+2) & \ldots & x(n) \end{bmatrix} \quad (4)$$

In this method, the trajectory matrix plays an important role in defining the relationship between the window length $m$ and the total number of data points $n$. Typically, the window length is chosen as one quarter of the data. If the data is periodic, the window length can also be chosen as one quarter of the longest period in the data. Singular value decomposition of the trajectory matrix $X$ yields three matrices $U$, $W$, and $V$, expressed as:

$$X = UWV^T \quad (5)$$

where $U$ and $V$ are regular orthogonal matrices, and $W$ is a diagonal matrix that describes the diagonal components of the singular values $\lambda_i$. $U = (u_1, u_2, ..., u_r)$, $V = (v_1, v_2, ..., v_r)$ and $r \leq \min(m, n)$, $X$ can be decomposed into the sum of several matrices as follows:

$$X = X_1 + X_2 + ... + X_r \quad (6)$$

$$X_i = \sqrt{\lambda_i} U_i V_i^T \ (i = 1, 2, ..., r) \quad (7)$$

The singular values of these matrices decrease with increasing subscripts, indicating that modes with smaller mode number have larger variances. Therefore, modes with smaller subscripts contribute more to the original signal. By eliminating signals with relatively small partial correlation, the remaining signals can be reconstructed to obtain the variable components that do not include irrelevant signals.By using the reconfigured BVP signal from $X$, a robust heart rate estimation can be obtained. We use a weight-correlated method to set the threshold, ensuring that noise can be effectively eliminated without losing the original valid signals.We performed a fast Fourier transform (FFT) on the BVP signal to convert it to the frequency domain and analyzed its power spectral density (PSD). Since the heart rate signal appears as a distinct peak in the frequency spectrum, we can approximate the heart rate value $f_{HR}$ by taking the frequency with the maximum spectral density:

$$f_{HR} = argmax|W(f)| \quad (8)$$

where $W(f)$ is the power spectral density of the BVP signal. Finally, we can obtain an estimated value of the heart rate, $HR$, is: $HR = f_{HR}*60$.

## 4  EXPERIMENTAL SETUP AND RESULTS

Due to the fact that the experiment was conducted in a far-field environment, there are currently no publicly available datasets to use. Therefore, in order to collect the necessary data, 13 subjects were self-collected, each video is 70 seconds long, including 10 males and 3 females. In this experiment, we used a network camera (Guke, G06-18X) to record the video. The camera was placed 5 meters away from the subjects, with a frame rate of 30 fps and a resolution of 640×480px. In order to produce the phenomenon of illumination changes, we placed a white LED light panel (FengChuan Ltd., Shenzhen, China) with 18W power at a distance of 1.5 meters from the

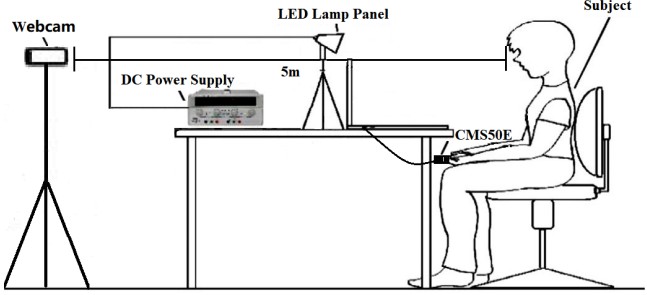

Figure 4: The scene diagram of the experiment.

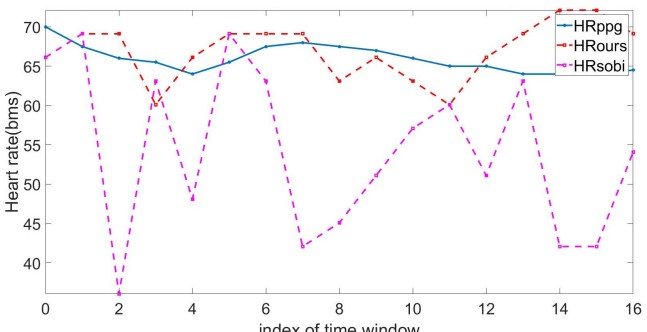

Figure 5: Changes of heart rate at different times between the device, SOBI algorithm and the proposed algorithm.

subjects. In order to obtain the ground truth of the subject's heart rate, we used a finger clip oximeter (ContecMedice, CMS50E) to measure the photoplethysmogram (PPG) on the subjects' fingers. The scene of the experiment is shown in Fig. 4.

## 5  RESULT ANALYSIS

In this section, we used different denoising methods to process the raw signal.In order to evaluate the performance of different denoising methods, we compared the actual heart rate measured by the finger clip oximeter with the heart rate obtained after denoising. In order to assess the accuracy of heart rate measurement, we used two metrics: mean absolute error (MAE) and root mean square error (RMSE). The formulas for calculating RMSE and MAE are as follows:

$$RMSE = \sqrt{\frac{1}{n}\sum_{i=1}^{n}(h_i(x) - y_i)^2} \quad (9)$$

$$MAE = \frac{1}{n}\sum_{i=1}^{n}|h_i(x) - y_i| \quad (10)$$

where $n$ is the number of measurements, $y$ represents the true heart rate value, and $h(x)$ represents ground truth HR obtained by the method. The smaller the values of RMSE and MAE, the better the denoising performance. By calculating these metrics, we can evaluate the advantages and disadvantages of different denoising methods and find the best denoising method to provide more accurate results for heart rate measurement.

We took an example video from our own collected database to analyze the changes of heart rate obtained by using the standard PPG signal collected by the CMS50E device, the heart rate obtained by using the SOBI denoising algorithm, and the heart rate obtained by our proposed method at different times, as shown in Fig. 5.In Fig. 5, $HR_{ppg}$ represents the ground truth HR collected by the instrument, $HR_{ours}$ is the heart rate obtained using our proposed method, and

Table 1: The RMSE and MAE values of generally selected ROI under different methods.

| method | G-BVP | SOBI | OURS |
|---|---|---|---|
| RMSE(bpm) | 18.23 | 13.71 | 9.52 |
| MAE(bpm) | 16.17 | 11.74 | 7.76 |

Table 2: The RMSE and MAE values of feature point tracked ROI under different methods.

| method | G-BVP | SOBI | OURS |
|---|---|---|---|
| RMSE(bpm) | 18.05 | 12.96 | 8.9 |
| MAE(bpm) | 15.95 | 10.62 | 7.13 |

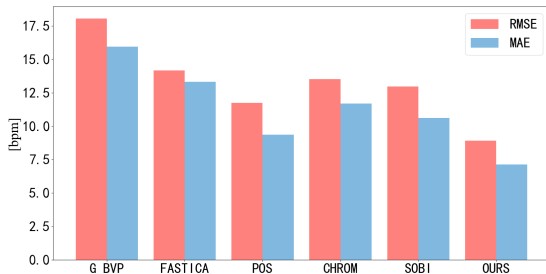

Figure 7: RMSE and MAE of various classical methods and proposed methods.

$HR_{sobi}$ is the heart rate obtained using the SOBI algorithm. It can be seen that in the scenario with changing illumination, the heart rate measurement obtained directly using the SOBI algorithm is not particularly accurate. In contrast, our proposed method can more accurately fit the heart rate measured by the instrument.

We compared generally selecting regions of interest with using feature point tracking to define regions of interest, several metrics were used to demonstrate the experiment. To present the results more intuitively, we used Table 1 and Table 2 to list their RMSE and MAE values.From Table 1 and Table 2, the difference between generally selecting ROI and using feature point tracking ROI is not very significant, but it can still be seen that the feature point tracking method is slightly better than the general selection method. In the experiment, the RMSE of the G-BVP method with general selection and feature point tracking were 18.23bpm and 18.05bpm, respectively, and their MAE were 16.17bpm and 15.95bpm, respectively. The RMSE of the SOBI algorithm were 13.71bpm and 12.96bpm, respectively, and their MAE were 11.74bpm and 10.62bpm, respectively. The proposed method had RMSE of 9.52bpm and 8.9bpm, respectively, and their MAE were 7.76bpm and 7.13bpm, respectively. Feature point tracking has smaller errors. The feature point tracking algorithm can automatically identify the position of the ROI and adaptively track the ROI, which can better cope with the interference caused by facial expressions and other factors, and obtain more accurate heart rate measurement results.

To verify the performance of the method, We used some typical methods to estimate heart rate using videos in the self-collecting database, These methods including G-BVP, FastICA [6] , POS, CHROM, and SOBI. We compared them with the proposed method to validate the experimental results.Fig. 6 shows the heart rate range plot drawn based on the results obtained from different methods. The red line represents the average value, the blue squares represent the distribution range of most values, and the red dots represent individual outlier values. Referring to the values of PPG, it can be seen that our proposed method is basically consistent with the range of PPG, while other methods may have large value range offsets or more outlier points, and the results obtained are not accurate enough. At the same time, the average values of the POS algorithm and our proposed method are basically consistent with the values measured by the instrument.

It can be seen from Fig. 7 that the proposed method has the smallest RMSE and MAE. it can be seen that compared with our proposed method, the RMSE and MAE obtained by other methods are relatively large. The POS algorithm has the best performance among other methods, followed by SOBI, CHROM, FastICA, and G-BVP. The RMSE of our proposed method is 8.9 bpm, and the MAE is 7.13 bpm, which also indicates that the proposed method has strong resistance to changes in illumination.

In the case of remote measurement in the far field with changing illumination, the statistical values of all remote measurement methods based on the Bland-Altman method are shown in Fig. 8. According to G-BVP, as shown in Fig. 8(a), the average deviation and consistency range are -10, -40.32+20.31 beats/min. According to FastICA, as shown in Figure Fig. 8(b), the average deviation and consistency range are -12.16, -30.9+14.1 beats/min. According to POS, as shown in Figure Fig. 8(c), the average deviation and consistency range are -1.67, -26.9+23.57 beats/min. According to CHROM, as shown in Figure Fig. 8(d), the average deviation and consistency range are -3.53, -34.77+27.72 beats/min. According to SOBI, as shown in Fig. 8(e), the average deviation and consistency range are -8.14 beats/min, -30.9+14.1 beats/min. Using our proposed framework, as shown in Fig. 8(f), the average deviation and consistency range are -3.17, -18.82+12.49 beats/min. Based on the statistical results in Figure 10, it can be seen that POS, CHROM, and our proposed method have smaller average offsets, among which our proposed method has the smallest standard deviation and almost no outliers deviating from the consistency range. This indicates that our proposed method has good performance in remote heart rate measurement.

## 6 CONCLUSION

Our research not only extends the distance for measuring heart rate, but also proposes a new framework for remote heart rate measurement. In response to the performance degradation of previous common face video remote heart rate measurement methods under environmental illumination changes and subject movement interference, we propose a framework consisting of three main processes. Firstly, we use the VJ algorithm to accurately identify the ROI of the face and solve the problem caused by rigid head movement. At the same time, we use KLT technology to continuously track each frame to further reduce motion interference. Secondly, we use FWT to remove initial illumination interference, and then use the SOBI algorithm to remove most of the noise. Finally, we use the SSA method to remove residual motion artifact noise. We conducted

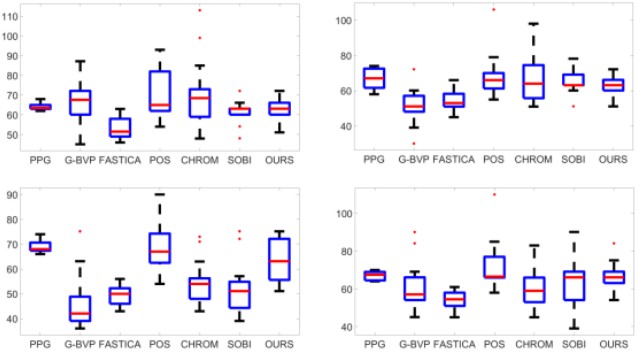

Figure 6: Heart rate ranges and outlier distributions of different methods.

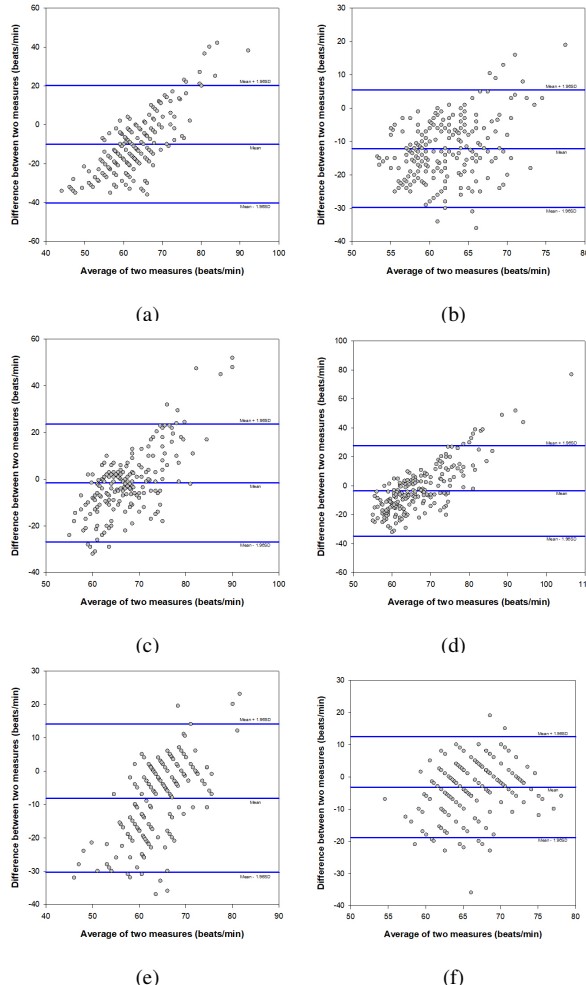

Figure 8: Bland-Altman plots. (a) G-BVP method, (b) FastICA method, (c) POS method, (d) CHROM method, (e) SOBI method, (f) Ours method.

extensive experimental evaluations of this method and compared it with reference measurement results. The results show that in a far-field environment, this framework can accurately measure the heart rate of people under interference of illumination. We evaluated the system using multiple video data sources and found that the system exhibits strong consistency, high correlation, and low noise levels. Moreover, in complex and changing situations, the system's results are better than traditional measurement methods such as G-BVP, FastICA, POS, CHROM, and SOBI algorithms. Our research results have important implications for the development of remote heart rate measurement technology. Our method not only extends the measurement distance, but also reduces the interference of environmental illumination changes and subject movement on measurement results, thereby improving the practicality and reliability of this technology and expanding the application scenarios of rPPG. We believe that in the future, this technology will be widely used to provide more convenient and accurate monitoring means for people's health.

### ACKNOWLEDGMENTS

I would like to express my sincere gratitude to my supervisor, Chao Zhang, for his invaluable guidance, support and encouragement throughout my research work. His extensive knowledge, critical insights and constructive feedback have been instrumental in shaping my research and helping me achieve my academic goals. I am also deeply grateful to him for his patience, kindness and understanding, which have made my research journey a truly enriching and rewarding experience. I will always cherish the lessons I have learned from him and the memories we have shared together. Thank you, Chao Zhang, for everything you have done for me.

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
