# OpenReview forum: "A Non-Contact Heart Rate Measurement Method Resistant to Illumination Changes Based on Fast Wavelet Transform and Second-Order Blind Identification in Far-Field Environments"
_graphicsinterface.org/Graphics_Interface/2023/Conference_SD — Submitted to GI 2023 - second deadline_

### Official Review · Reviewer_eSXp · 2023-04-21
**no anonymity**

**Rating:** 3
**Confidence:** 1

**Review:**

This paper breaks the double blind review process. The author names are visible.

I am not an expert at all in Heart Rate Measurement at all and do not feel comfortable giving an export review.

---

### Official Review · Reviewer_2RJ2 · 2023-04-23
**Rejection due to a lack of scientific contribution in computer vision, HCI or computer graphics**

**Rating:** 3
**Confidence:** 5

**Review:**

This paper presents a method to detect heart rate from video analysis. The techniques are limited to standard image processing approaches. Compared to state of the art in Computer Vision, this paper brings very limited scientific contribution. Validation section is very vague and does not reach the requirements expected in HCI. To notice, the authors should be aware that the submission must be anonymous, with no authors name or acknowledgment section.

---

### Official Review · Reviewer_Equv · 2023-04-24
**Out-of-scope and straightforward**

**Rating:** 2
**Confidence:** 3

**Review:**

The paper includes author information, which violates the double-blind policy. Nevertheless, I provide my comments below.

The paper presents a remote photoplethysmography (rPPG) framework for heart rate detection in far-field environments. The proposed method addresses the limitations of existing close-range heart rate measurement techniques by selecting a larger region of interest (ROI) using feature point tracking.


The authors tackle an important problem -of remote heart rate measurement in far-field environments-, however, GI might not be the right venue for publication. Furthermore, the proposed method is straightforward and relies on well-established techniques. It is unclear where the novelty lies. I also find the experimentation and comparisons to be quite limited.